

# A novel CAPTCHA solver framework using deep skipping Convolutional Neural Networks

Shida Lu[1], Kai Huang[2], Talha Meraj[3] and Hafiz Tayyab Rauf[4]

[1] State Grid Information & Communication Company, SMEPC, Shanghai, China
[2] Shanghai Shineenergy Information Technology Development Co., Ltd., Shanghai, China
[3] COMSATS Institute of Information Technology, Islamabad, Pakistan
[4] University of Bradford, Bradford, United Kingdom

## ABSTRACT

A Completely Automated Public Turing Test to tell Computers and Humans Apart (CAPTCHA) is used in web systems to secure authentication purposes; it may break using Optical Character Recognition (OCR) type methods. CAPTCHA breakers make web systems highly insecure. However, several techniques to break CAPTCHA suggest CAPTCHA designers about their designed CAPTCHA's need improvement to prevent computer vision-based malicious attacks. This research primarily used deep learning methods to break state-of-the-art CAPTCHA codes; however, the validation scheme and conventional Convolutional Neural Network (CNN) design still need more confident validation and multi-aspect covering feature schemes. Several public datasets are available of text-based CAPTCHa, including Kaggle and other dataset repositories where self-generation of CAPTCHA datasets are available. The previous studies are dataset-specific only and cannot perform well on other CAPTCHA's. Therefore, the proposed study uses two publicly available datasets of 4- and 5-character text-based CAPTCHA images to propose a CAPTCHA solver. Furthermore, the proposed study used a skip-connection-based CNN model to solve a CAPTCHA. The proposed research employed 5-folds on data that delivers 10 different CNN models on two datasets with promising results compared to the other studies.

## INTRODUCTION

The first secure and fully automated mechanism, named CAPTCHA, was developed in 2000. Alta Vista first used the term CAPTCHA in 1997. It reduces spamming by 95% (*Baird & Popat, 2002*). CAPTCHA is also known as a reverse Turing test. The Turing test was the first test to distinguish human, and machine (*Von Ahn et al., 2003*). It was developed to determine whether a user was a human or a machine. It increases efficiency against different attacks that seek websites (*Danchev, 2014*; *Obimbo, Halligan & De Freitas, 2013*). It is said that CAPTCHA should be generic such that any human can easily interpret and solve it and difficult for machines to recognize it (*Bostik & Klecka, 2018*). To protect against robust malicious attacks, various security authentication methods have been

Corresponding author
Shida Lu, lushida621@163.com

developed (*Goswami et al., 2014*; *Priya & Karthik, 2013*; *Azad & Jain, 2013*). CAPTCHA can be used for authentication in login forms, spam text reducer, *e.g.*, in email, as a secret graphical key to log in for email. In this way, a spam-bot would not be able to recognize and log in to the email (*Sudarshan Soni & Bonde, 2017*). However, recent advancements make the CAPTCHA's designs to be a high risk where the current gaps and robustness of models that are the concern is discussed in depth (*Roshanbin & Miller, 2013*). Similarly, the image, text, colorful CAPTCHAs, and other types of CAPTCHA's are being attacked by various malicious attacks. However, most of them have used deep learning based methods to crack them due to their robustness and confidence (*Xu, Liu & Li, 2020*).

Many prevention strategies against malicious attacks have been adopted in recent years, such as cloud computing-based voice-processing (*Gao, Wang & Shen, 2020b*; *Gao, Wang & Shen, 2020a*), mathematical and logical puzzles, and text and image recognition tasks (*Gao, Wang & Shen, 2020c*). Text-based authentication methods are mostly used due to their easier interpretation, and implementation (*Madar, Kumar & Ramakrishna, 2017*; *Gheisari et al., 2021*). A set of rules may define a kind of automated creation of CAPTCHA-solving tasks. It leads to easy API creation and usage for security web developers to make more mature CAPTCHAs (*Bursztein et al., 2014*; *Cruz-Perez et al., 2012*). The text-based CAPTCHA is used for Optical Character Recognition (OCR). OCR is strong enough to solve text-based CAPTCHA challenges. However, it still has challenges regarding its robustness in solving CAPTCHA problems (*Kaur & Behal, 2015*). These CAPTCHA challenges are extensive with ongoing modern technologies. Machines can solve them, but humans cannot. These automated, complex CAPTCHA-creating tools can be broken down using various OCR techniques. Some studies claim that they can break any CAPTCHA with high efficiency. The existing work also recommends strategies to increase the keyword size and another method of crossing lines from keywords that use only straight lines and a horizontal direction. It can break easily using different transformations, such as the Hough transformation. It has also been suggested that single-character recognition is used from various angles, rotations, and views to make more robust and challenging CAPTCHAs (*Bursztein, Martin & Mitchell, 2011*).

The concept of reCAPTCHA was introduced in 2008. It was initially a rough estimation. It was later improved and was owned by Google to decrease the time taken to solve it. The unsolvable reCAPTCHAs were then considered to be a new challenge for OCRs (*Von Ahn et al., 2008*). The usage of computer vision and image processing as a CAPTCHA solver or breaker was increased if segmentation was performed efficiently (*George et al., 2017*; *Ye et al., 2018*). The main objective or purpose of making a CAPTCHA solver is to protect CAPTCHA breakers. By looking into CAPTCHA solvers, more challenging CAPTCHAs can be generated, and they may lead to a more secure web that is protected against malicious attacks (*Kumar, 2021*). A benchmark or suggestion for CAPTCHA creation was given by Chellapilla et al.: Humans should solve the given CAPTCHA challenge with a 90% success rate, while machines ideally solve only one in every 10,000 CAPTCHAs (*Chellapilla et al., 2005*).

Modern AI yields CAPTCHAs that can solve problems in a few seconds. Therefore, creating CAPTCHAs that are easily interpretable for humans and unsolvable for machines

is an open challenge. It is also observed that humans invest a substantial amount of time daily solving CAPTCHAs (*Von Ahn et al., 2008*). Therefore, reducing the amount of time humans need to solve them is another challenge. Various considerations need to be made, including text familiarity, visual appearance, distortions, *etc.* Commonly in text-based CAPTCHAs, the well-recognized languages are used that have many dictionaries that make them easily breakable. Therefore, we may need to make unfamiliar text from common languages such as phonetic text is not ordinary language that is pronounceable (*Wang & Bentley, 2006*). Similarly, the color of the foreground and the background of CAPTCHA images is also an essential factor, as many people have low or normal eyesight or may not see them. Therefore, a visually appealing foreground and background with distinguishing colors are recommended when creating CAPTCHAs. Distortions from periodic or random manners, such as affine transformations, scaling, and the rotation of specific angles, are needed. These distortions are solvable for computers and humans. If the CAPTCHAs become unsolvable, then multiple attempts by a user are needed to read and solve them (*Yan & El Ahmad, 2008*).

In current times, Deep Convolutional neural networks (DCNN) are used in many medical (*Meraj et al., 2019*; *Manzoor et al., 2022*; *Mahum et al., 2021*) and other real-life recognition applications (*Namasudra, 2020*) as well as insecurity threat solutions (*Lal et al., 2021*). The security threats in IoT and many other aspects can also be controlled using blockchain methods (*Namasudra et al., 2021*). Utilizing deep learning, the proposed study uses various image processing operations to normalize text-based image datasets. After normalizing the data, a single-word-caption-based OCR was designed with skipping connections. These skipping connections connect previous pictorial information to various outputs in simple Convolutional Neural Networks (CNNs), which possess visual information in the next layer only (*Ahn & Yim, 2020*).

The main contribution of this research work is as follows:

- A skipping-connection-based CNN framework is proposed that covers multiple aspects of features.
- A 5-fold validation scheme is used in a deep-learning-based network to remove bias, if any, which leads to more promising results.
- The data are normalized using various image processing steps to make it more understandable for the deep learning model.

## LITERATURE REVIEW

Today in the growing and dominant field of AI, many real-life problems have been solved with the help of deep learning and other evolutionary optimized intelligent algorithms (*Rauf, Bangyal & Lali, 2021*; *Rauf et al., 2020*). Various problems of different aspects using DL methods are solved, such as energy consumption analysis (*Gao, Wang & Shen, 2020b*), time scheduling of resources to avoid time and resources wastage (*Gao, Wang & Shen, 2020c*). Similarly, in cybersecurity, a CAPTCHA solver has provided many automated AI solutions, except OCR. Multiple proposed CNN models have used various types of CAPTCHA datasets to solve CAPTCHAs. The collected datasets have been divided

into three categories: selection-, slide-, and click-based. Ten famous CAPTCHAs were collected from google.com, tencent.com, etc. The breaking rate of these CAPTCHAs was compared. CAPTCHA design flaws that may help to break CAPTCHAs easily were also investigated. The underground market used to solve CAPTCHAs was also investigated, and findings concerning scale, the commercial sizing of keywords, and their impact on CAPTCHAs were reported (*Weng et al., 2019*). A proposed sparsity-integrated CNN used constraints to deactivate the fully connected connections in CNN. It ultimately increased the accuracy results compared to transfer learning, and simple CNN solutions (*Ferreira et al., 2019*).

Image processing operations regarding erosion, binarization, and smoothing filters were performed for data normalization, where adhesion-character-based features were introduced and fed to a neural network for character recognition (*Hua & Guoqin, 2017*). The backpropagation method was claimed as a better approach for image-based CAPTCHA recognition. It has also been said that CAPTCHA has become the normal, secure authentication method in the majority of websites and that image-based CAPTCHAs are more valuable than text-based CAPTCHAs (*Saroha & Gill, 2021*). Template-based matching is performed to solve text-based CAPTCHAs, and preprocessing is also performed using Hough transformation and skeletonization. Features based on edge points are also extracted, and the points of reference with the most potential are taken. It is also claimed that the extracted features are invariant to position, language, and shapes. Therefore, it can be used for any merged, rotated, and other variation-based CAPTCHAs (*Wang, 2017*).

PayPal CAPTCHAs have been solved using correlation, and Principal Component Analysis (PCA) approaches. The primary steps of these studies include preprocessing, segmentation, and the recognition of characters. A success rate of 90% was reported using correlation analysis of PCA and using PCA only increased the efficiency to 97% (*Rathoura & Bhatiab, 2018*). A Faster Recurrent Neural Network (F-RNN) has been proposed to detect CAPTCHAs. It was suggested that the depth of a network could increase the mean average precision value of CAPTCHA solvers, and experimental results showed that feature maps of a network could be obtained from convolutional layers (*Du et al., 2017*). Data creation and cracking have also been used in some studies. For visually impaired people, there should be solutions to CAPTCHAs. A CNN network named CAPTCHANet has been proposed.

A 10-layer network was designed and was improved later with training strategies. A new CAPTCHA using Chinese characters was also created, and it removed the imbalance issue of class for model training. A statistical evaluation led to a higher success rate (*Zhang et al., 2021*). A data selection approach automatically selected data for training purposes. The data augmenter later created four types of noise to make CAPTCHAs difficult for machines to break. However, the reported results showed that, in combination with the proposed preprocessing method, the results were improved to 5.69% (*Che et al., 2021*). Some recent studies on CAPTCHA recognition are shown in Table 1.

The pre-trained model of object recognition has an excellent structural CNN. A similar study used a well-known VGG network and improved the structure using focal loss

**Table 1   Few recent CAPTCHA recognition-based studies methods and their results.**

| Reference | Year | Dataset | Method | Results |
|---|---|---|---|---|
| *Wang & Shi (2021)* | 2021 | CNKI CAPTCHA, Random Generated, Zhengfang CAPTCHA | Binarization, smoothing, segmentation and annotation with Adhesian and more interference | Recognition rate = 99%, 98.5%, 97.84% |
| *Ahmed & Anand (2021)* | 2021 | Tamil, Hindi and Bengali | Pillow Library, CNN | ~ |
| *Bostik et al. (2021)* | 2021 | Private created Dataset | 15-layer CNN | Classification accuracy = 80% |
| *Kumar & Singh (2021)* | 2021 | Private | 7-Layer CNN | Classification Accuracy = 99.7% |
| *Dankwa & Yang (2021)* | 2021 | 4-words Kaggle Dataset | CNN | Classification Accuracy = 100% |
| *Wang et al. (2021)* | 2021 | Private GAN based dataset | CNN | Classification Accuracy = 96%, overall = 74% |
| *Thobhani et al. (2020)* | 2020 | Weibo, Gregwar | CNN | Testing Accuracy = 92.68% Testing Accuracy = 54.20% |

(*Wang & Shi, 2021*). The image processing operations generated complex data in text-based CAPTCHAs, but there may be a high risk of breaking CAPTCHAs using common languages. One study used the Python Pillow library to create Bengali-, Tamil-, and Hindi-language-based CAPTCHAs. These language-based CAPTCHAs were solved using D-CNN, which proved that the model was also confined by these three languages (*Ahmed & Anand, 2021*). A new, automatic CAPTCHA creating and solving technique using a simple 15-layer CNN was proposed to remove the manual annotation problem.

Various fine-tuning techniques have been used to break 5-digit CAPTCHAs and have achieved 80% classification accuracies (*Bostik et al., 2021*). A privately collected dataset was used in a CNN approach with seven layers that utilize correlated features of text-based CAPTCHAs. It achieved a 99.7% accuracy using its image database, and CNN architecture (*Kumar & Singh, 2021*). Another similar approach was based on handwritten digit recognition. The introduction of a CNN was initially discussed, and a CNN was proposed for twisted and noise-added CAPTCHA images (*Cao, 2021*). A deep, separable CNN for four-word CAPTCHA recognition achieved 100% accurate results with the fine-tuning of a separable CNN concerning their depth. A fine-tuned, pre-trained model architecture was used with the proposed architecture and significantly reduced the training parameters with increased efficiency (*Dankwa & Yang, 2021*).

A visual-reasoning CAPTCHA (known as a Visual Turing Test (VTT)) has been used in security authentication methods, and it is easy to break using holistic and modular attacks. One study focused on a visual-reasoning CAPTCHA and showed an accuracy of 67.3% against holistic CAPTCHAs and an accuracy of 88% against VTT CAPTCHAs. Future

directions were to design VTT CAPTCHAs to protect against these malicious attacks (*Gao et al., 2021*). To provide a more secure system in text-based CAPTCHAs, a CAPTCHA defense algorithm was proposed. It used a multi-character CAPTCHA generator using an adversarial perturbation method. The reported results showed that complex CAPTCHA generation reduces the accuracy of CAPTCHA breaker up to 0.06% (*Wang, Zhao & Liu, 2021*). The Generative Adversarial Network (GAN) based simplification of CAPTCHA images adopted before segmentation and classification. A CAPTCHA solver is presented that achieves 96% success rate character recognition. All other CAPTCHA schemes were evaluated and showed a 74% recognition rate. These suggestions for CAPTCHA designers may lead to improved CAPTCHA generation (*Wang et al., 2021*). A binary image-based CAPTCHA recognition framework is proposed to generate a certain number of image copies from a given CAPTCHA image to train a CNN model. The Weibo dataset showed that the four-character recognition accuracy on the testing set was 92.68%, and the Gregwar dataset achieved a 54.20% accuracy on the testing set (*Thobhani et al., 2020*).

The reCAPTCHA images are a specific type of security layer used by some sites and set a benchmark by Google to meet their broken challenges. This kind of image would deliver specific images, and then humans have to pick up any similar image that could be realized by humans efficiently. The machine learning-based studies also discuss and work on these kinds of CAPTCHA images (*Alqahtani & Alsulaiman, 2020*). Drag and drop image CAPTCHA-based security schemes are also applied. An inevitable part of the image is missed and needs to be dragged, and the blank filled in in a particular location and shape. However, it could also be broken by finding space areas using neighborhood differences of pixels. Regardless, it is not good enough to avoid any malicious attacks (*Ouyang et al., 2021*).

Adversarial attacks are the rising challenge to deceive the deep learning models nowadays. To prevent deep learning model-based CAPTCHA attacks, many different adversarial noises are being introduced and used in security questions that create similar images. It needs to be found by the user. A sample image-based noise-images are generated and shown in the puzzle that could be found by human-eye with keen intention (*Shi et al., 2021*; *Osadchy et al., 2017*). However, these studies need self-work because noise-generated images can consume more time for users. Also, some of the adversarial noise-generating methods could generate unsolvable samples for some of the real-time users.

The studies discussed above yield information about text-based CAPTCHAs as well as other types of CAPTCHAs. Most studies used DL methods to break CAPTCHAs, and time and unsolvable CAPTCHAs are still an open challenge. More efficient DL methods need to be used that, though they may not cover other datasets, should be robust to them. The locally developed datasets are used by many of the studies make the proposed studies less robust. However, publicly available datasets could be used so that they could provide more robust and confident solutions.

## METHODOLOGY

Recent studies based on deep learning have shown excellent results to solve a CAPTCHA. However, simple CNN approaches may detect lossy pooled incoming features when passing
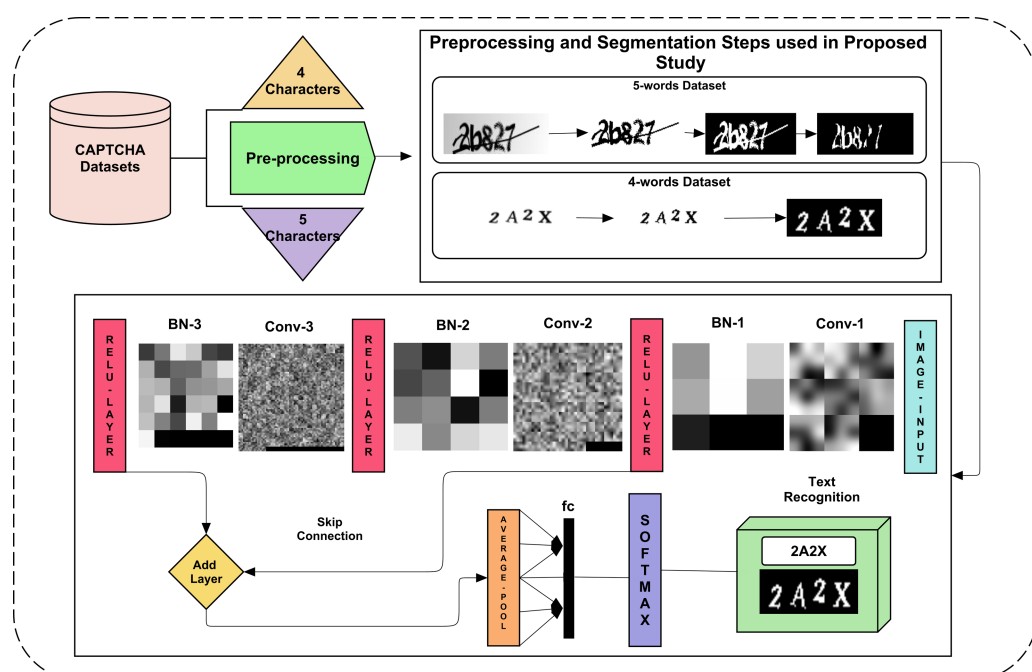

**Figure 1** The proposed framework for CAPTCHA recognition for both 4 and 5 character datasets.

between convolution and other pooling layers. Therefore, the proposed study utilizes skip connection. To remove further bias, a 5-fold validation approach is adopted. The proposed study presents a CAPTCHA solver framework using various steps, as shown in Fig. 1. The data are normalized using various image processing steps to make it more understandable for the deep learning model. This normalized data is segmented per character to make an OCR-type deep learning model that can detect each character from each aspect. At last, the five-fold validation method is reported and yields promising results.

The two datasets used for CAPTCHA recognition have four and five words in them. The five-word dataset has a horizontal line in it with overlapping text. Segmenting and recognizing such text is challenging due to its un-clearance. The other dataset of 4 characters was not as challenging to segment, as no line intersected them, and character rotation scaling needs to be considered. Their preprocessing and segmentation are explained in the next section. The dataset is explored in detail before and after preprocessing and segmentation.

## Datasets

There are two public datasets available on Kaggle that are used in the proposed study. There are five and four characters in both datasets. There are different numbers of numeric and alphabetic characters in them. There are 1,040 images in the five-character dataset ($d_1$) and 9,955 images in the four-character dataset ($d_2$). There are 19 types of characters in the $d_1$ dataset, and there are 32 types of characters in the $d_2$ dataset. Their respective dimensions and extension details before and after segmentation are shown in Table 2. The frequencies of each character in both datasets are shown in Fig. 2.

**Table 2  Description of both employed datasets' in proposed study.**

| Properties | d1 | d2 |
|---|---|---|
| Image dimension | $50 \times 200 \times 3$ | $24 \times 72 \times 3$ |
| Extension | PNG | PNG |
| Number of images | 9955 | 1040 |
| Character types | 32 | 19 |
| Resized image dimension (Per Character) | $20 \times 24 \times 1$ | $20 \times 24 \times 1$ |

The frequency of each character varies in both datasets, and the number of characters also varies. In the $d_2$ dataset, although there is no complex inner line intersection and a merging of texts is found, more characters and their frequencies are. However, the $d_1$ dataset has complex data and a low number of characters and frequencies, as compared to $d_2$. Initially, $d_1$ has the dimensions $50 \times 200 \times 3$, where 50 represents the rows, 200 represents the columns, and 3 represents the color depth of the given images. $d_2$ has image dimensions of $24 \times 72 \times 3$, where 24 is the rows, 72 is the columns, and 3 is the color depth of given images. These datasets have almost the same character location. Therefore, they can be manually cropped to train the model on each character in an isolated form. However, their dimensions may vary for each character, which may need to be equally resized. The input images of both datasets were in Portable Graphic Format (PNG) and did not need to change. After segmenting both dataset images, each character is resized to $20 \times 24$ in both datasets. This size covers each aspect of the visual binary patterns of each character. The dataset details before and after resizing are shown in Table 2.

The summarized details of the used datasets in the proposed study are shown in Table 2. The dimensions of the resized image per character mean that, when we segment the characters from the given dataset images, their sizes vary from dataset to dataset and from character type to character type. Therefore, the optimal size at which the image data for each character is not lost is 20 rows by 24 columns, which is set for each character.

## Preprocessing and segmentation

$d_1$ dataset images do not need any complex image processing to segment them into a normalized form. $d_2$ needs this operation to remove the central intersecting line of each character. This dataset can be normalized to isolate each character correctly. Therefore, three steps are performed on the $d_1$ dataset. It is firstly converted to greyscale; it is then converted to a binary form, and their complement is lastly taken. In the $d_2$ dataset, 2 additional steps of erosion and area-wise selection are performed to remove the intersection line and the edges of characters. The primary steps of both datasets and each character isolation are shown in Fig. 3.

Binarization is the most needed step in order to understand the structural morphology of a certain character in a given image. Therefore, greyscale conversion of images is performed to perform binarization, and images are converted from greyscale to a binary format. The RGB format image has 3 channels in them: Red, Green, and Blue. Let Image $I_{(x,y)}$ be the input RGB image, as shown in Eq. (1). To convert these input images into grayscale, Eq. (2)

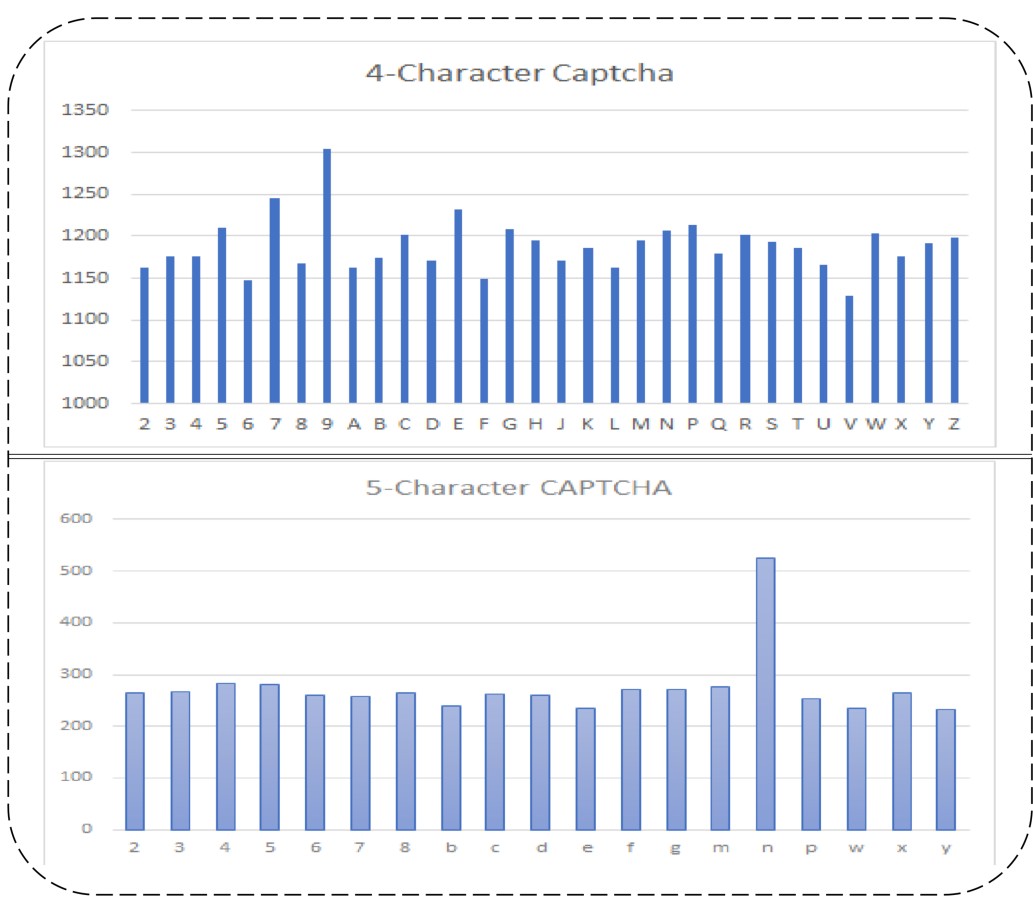

**Figure 2 Five and four character's datasets used in proposed study, their character-wise frequencies (row 1: 4-character dataset 1 ($d_2$); row 2: five-character Dataset 2 ($d_1$)).**

is performed.

$$Input\ Image = I_{(x,y)}. \tag{1}$$

In Eq. (1), $I$ is the given image, and x and y $x$ *and* $y$ represent the rows and columns. The grayscale conversion is performed using Eq. (2):

$$Grey\ (x,y) \leftarrow \sum_{i=n}^{j} (0.2989 * R,\ 0.5870 * G, 0.1140 * B). \tag{2}$$

In Eq. (2), $i$ is the iterating row position, $j$ is the interacting column position of the operating pixel at a certain time, and $R$, $G$, and $B$ are the red, green, and blue pixel values of that pixel. The multiplying constant values convert to all three values of the respective channels to a new grey-level value in the range of 0–255. $Grey\ (x,y)$ is the output grey-level of a given pixel at a certain iteration. After converting to grey-level, the binarization operation is performed using Bradly's method, which calculates a neighborhood base threshold to convert into 1 and 0 values to a given grey-level matrix of dimension 2. The

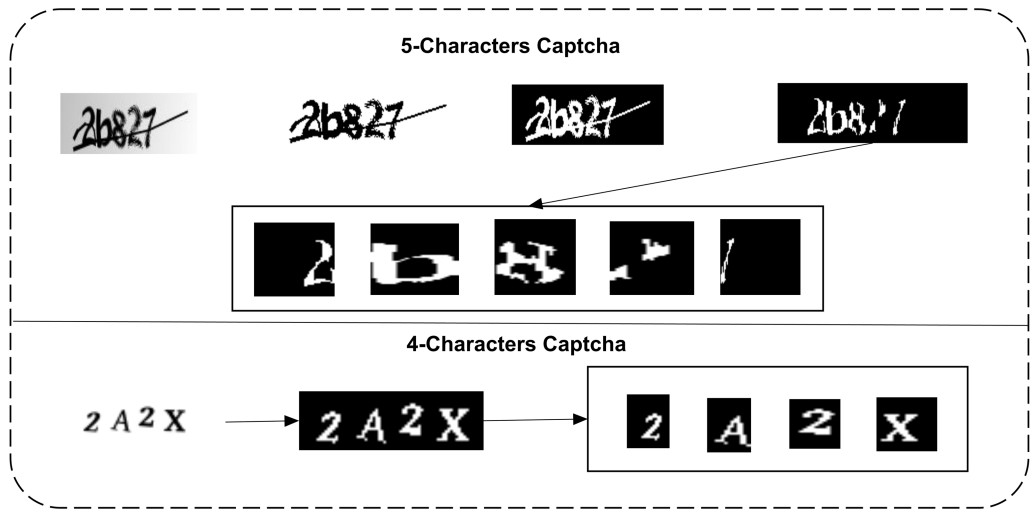

**Figure 3** Preprocessing and isolation of characters in both datasets (row 1: the d1 dataset, binarization, erosion, area-wise selection, and segmentation; row 2: binarization and isolation of each character).

neighborhood threshold operation is performed using Eq. (3).

$$B\left(x,y\right) \leftarrow 2 * \lfloor \text{size} \left( \frac{\text{Grey}\left(x,y\right)}{16} + 1\right)\rfloor. \tag{3}$$

In Eq. (3), the output $B\left(x,y\right)$ is the neighborhood-based threshold that is calculated as the 1/8 $^{th}$ neighborhood of a given $Grey\left(x,y\right)$ image. However, the floor is used to obtain a lower value to avoid any miscalculated threshold value. This calculated threshold is also called the adaptive threshold method. The neighborhood value can be changed to increase or decrease the binarization of a given image. After obtaining a binary image, the complement is necessary to highlight the object in a given image, taken as a simple inverse operation, calculated as shown in Eq. (4).

$$C\left(x,y\right) \leftarrow \frac{1}{B(x,y)}. \tag{4}$$

In Eq. (4), the available 0 and values are inverted to their respective values of each pixel position $x$ and $y$. The inverted image is used as an isolation process in the case of the $d_2$ dataset. In the case of the $d_1$, further erosion is needed. Erosion is an operation that uses a structuring element concerning its shape. The respective shape is used to remove pixels from a given binary image. In the case of a CAPTCHA image, the intersected line is removed using a line-type structuring element. The line-type structuring element uses a neighborhood operation. In the proposed study case, a line of size 5 with an angle dimension of 90 is used, and the intersecting line for each character in the binary image is removed, as we can see in Fig. 3, row 1. The erosion operation with respect to a 5 length and a 90 angle is calculated as shown in Eq. (5).

$$C \ominus L \leftarrow x \in E | B_x \subseteq C. \tag{5}$$

In Eq. (5), $C$ is the binary image, $L$ is the line type structuring element of line type, and $x$ is the resultant eroded matrix of the input binary image $C$. $B_x$ is the subset of a given image, as it is extracted from a given image $C$. After erosion, there is noise in some images that may lead to the wrong interpretation of that character. Therefore, to remove noise, the neighborhood operation is again utilized, and 8 neighborhood operations are used to a given threshold of 20 pixels for 1 value, as the noise value remains lower than the character in that binary image. To calculate it, an area calculation using each pixel is necessary. Therefore, by iterating an 8 by 8 neighborhood operation, 20 pixels consisting of the area are checked to remove those areas, and other more significant areas remain in the output image. The sum of a certain area with a maximum of 1 is calculated as shown in Eq. (6).

$$S(x,y) \leftarrow \sum_{i=1}^{j} \max(B_x|xi - xj|, B_x|yi - yj|). \tag{6}$$

In Eq. (6), the given rows ($i$) and columns ($j$) of a specific eroded image $B_x$ are used to calculate the resultant matrix by extracting each pixel value to obtain one's value from the binary image. The *max* will return only values that will be summed to obtain an area that will be compared with threshold value T. The noise will then be removed, and final isolation is performed to separate each normalized character.

## CNN training for text recognition

$$convo(I, W)_{x,y} = \sum_{a=1}^{N_C} \sum_{b=1}^{N_R} W_{a,b} * I_{x+a-1, y+b-1}. \tag{7}$$

In the above equation, we formulate a convolutional operation for a 2D image that represents $I_{x,y}$, where x and y are the rows and columns of the image, respectively. $W_{x,y}$ represents the convolving window concerning rows and columns $x$ and $y$. The window will iteratively be multiplied with the respective element of the given image and then return the resultant image in $convo(I, W)_{x,y}$. $N_C$ and $N_R$ are the numbers of rows and columns starting from 1, $a$ represents columns, and $b$ represents rows.

### Batch Normalization Layer

Its basic formula is to calculate a single component value, which can be represented as

$$Bat' = \frac{a - M[a]}{\sqrt{var(a)}}. \tag{8}$$

The calculated new value is represented as $Bat'$, a is any given input value, and $M[a]$ is the mean of that given value, where in the denominator the variance of input a is represented as $var(a)$. The further value is improved layer by layer to give a finalized normal value with the help of alpha gammas, as shown below:

$$Bat'' = \gamma * Bat' + \beta. \tag{9}$$

The extended batch normalization formulation improved in each layer with the previous $Bat'$ value.

### ReLU

ReLU excludes the input values that are negative and retains positive values. Its equation can be written as

$$reLU = \begin{Bmatrix} x = x \; if \; x > 0 \\ x = 0 \; if \; x \leq 0 \end{Bmatrix} \tag{10}$$

where x is the input value and directly outputs the value if it is greater than zero; if values are less than 0, negative values are replaced with 0.

### Skip-Connection

The Skip connection is basically concetnating the previous sort of pictoral information to the next convolved feature maps of network. In proposed network, the ReLU-1 information is saved and then after 2nd and 3rd ReLU layer, these saved information is concatenated with the help of an addition layer. In this way, the skip-connection is added that makes it different as compared to conventional deep learning approaches to classify the guava disease. Moreover, the visualization of these added feature information is shown in Fig. 1.

### Average pooling

The average pooling layer is superficial as we convolve to the input from the previous layer or node. The coming input is fitted using a window of size *mxn*, where m represents the rows, and n represents the column. The movement in the horizontal and vertical directions continues using stride parameters.

Many deep learning-based algorithms introduced previously, as we can see in Table 1, ultimately use CNN-based methods. However, all traditional CNN approaches using convolve blocks and transfer learning approaches may take important information when they pool down to incoming feature maps from previous layers. Similarly, the testing and validation using conventional training, validation, and testing may be biased due to less data testing than the training data. Therefore, the proposed study uses a 1-skip connection while maintaining other convolve blocks; inspired by the K-Fold validation method, it splits up both datasets' data into five respective folds. The dataset, after splitting into five folds, is trained and tested in a sequence. However, these five-fold results are taken as a means to report final accuracy results. The proposed CNN contains 16 layers in total, and it includes three major blocks containing convolutional, batch normalization, and ReLU layers. After these nine layers, an additional layer adds incoming connections, a skip connection, and 3rd-ReLU-layer inputs from the three respective blocks. Average pooling, fully connected, and softmax layers are added after skipping connections. All layer parameters and details are shown in Table 3.

In Table 3, all learnable weights of each layer are shown. For both datasets, output categories of characters are different. Therefore, in the dense layer of the five-fold CNN models, the output class was 19 for five models, and the output class was 32 categories in the other five models. The skip connection has more weights than other convolution layers. Each model is compared regarding its weight learning and is shown in Fig. 4.

The figure shows convolve 1, batch normalization, and skip connection weights. The internal layers have a more significant number of weights or learnable parameters, and the

**Table 3   Parameters setting and learnable weights for proposed skipping-CNN architecture.**

| Number | Layers name | Category | Parameters | Weights/Offset | Padding | Stride |
|---|---|---|---|---|---|---|
| 1 | Input | Image Input | $24 \times 20 \times 1$ | – | – | – |
| 2 | Conv (1) | Convolution | $24 \times 20 \times 8$ | $3 \times 3 \times 1 \times 8$ | Same | 1 |
| 3 | BN (1) | Batch Normalization | $24 \times 20 \times 8$ | $1 \times 1 \times 8$ | – | – |
| 4 | ReLU (1) | ReLU | $24 \times 20 \times 8$ | – | – | – |
| 5 | Conv (2) | Convolution | $12 \times 10 \times 16$ | $3 \times 3 \times 8 \times 16$ | Same | 2 |
| 6 | BN (2) | Batch Normalization | $12 \times 10 \times 16$ | $1 \times 1 \times 16$ | – | – |
| 7 | ReLU (2) | ReLU | $12 \times 10 \times 16$ | – | – | – |
| 8 | Conv (3) | Convolution | $12 \times 10 \times 32$ | $3 \times 3 \times 16 \times 32$ | Same | 1 |
| 9 | BN (3) | Batch Normalization | $12 \times 10 \times 32$ | $1 \times 1 \times 32$ | – | – |
| 10 | ReLU (3) | ReLU | $12 \times 10 \times 32$ | – | – | – |
| 11 | Skip-connection | Convolution | $12 \times 10 \times 32$ | $1 \times 1 \times 8 \times 32$ | 2 | 0 |
| 12 | Add | Addition | $12 \times 10 \times 32$ | – | – | – |
| 13 | Pool | Average Pooling | $6 \times 5 \times 32$ | – | 2 | 0 |
| 14 | FC | Fully connected | $1 \times 1 \times 19$ (d2) $1 \times 1 \times 32$ (d1) | $19 \times 960$ (d2) $32 \times 960$ (d1) | – | – |
| 15 | Softmax | Softmax | $1 \times 1 \times 19$ | – | – | – |
| 16 | Class Output | Classification | – | – | – | – |

different or contributing connection weights are shown in Fig. 4. Multiple types of feature maps are included in the figure. However, the weights of one dataset are shown. In the other dataset, these weights may vary slightly. The skip-connection weights have multiple features that are not in a simple convolve layer. Therefore, we can say that the proposed CNN architecture is a new way to learn multiple types of features compared to previous studies that use a traditional CNN. This connection may be used in other aspects of text and object recognition and classification.

Later on, by obtaining these significant, multiple features, the proposed study utilizes the K-fold validation technique by splitting the data into five splits. These multiple splits remove bias in the training and testing data and take the testing results as the mean of all models. In this way, no data will remain for training, and no data will be untested. The results ultimately become more confident than previous conventional approaches of CNN. The $d_2$ dataset has a clear, structured element in its segmented images; in $d_1$, the isolated text images were not much clearer. Therefore, the classification results remain lower in this case, whereas in the d2 dataset, the classification results remain high and usable as a CAPTCHA solver. The results of each character and dataset for each fold are discussed in the next section.

## RESULTS AND DISCUSSION

As discussed earlier, there are two datasets in the proposed framework. Both have a different number of categories and a different number of images. Therefore, separate evaluations of both are discussed and described in this section. Firstly, the five-character dataset is used by the 5-CNN models of same architecture, with a different split in the data. Secondly,

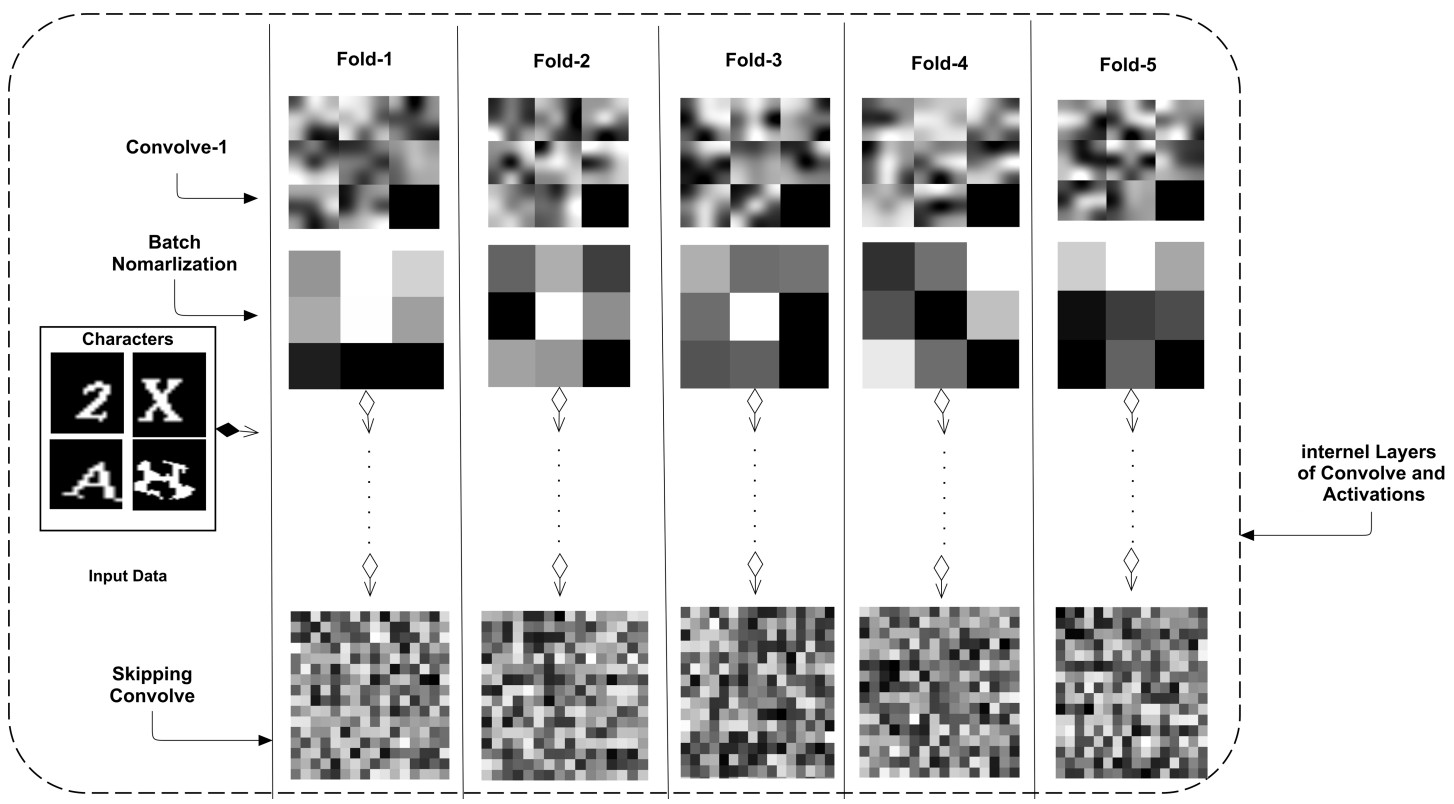

**Figure 4** Five-folds based trained CNN weights with their respective layers are shown that shows the proposed CNN skipping connection based variation in all CNNs' architectures.

the four-character dataset is used by the same architecture of the model, with a different output of classes.

## Five-character Dataset ($d_1$)

The five-character dataset has 1,040 images in it. After segmenting each type of character, it has 5,200 total images. The data are then split into five folds: 931, 941, 925, 937, and 924. The remaining data difference is adjusted into the training set, and splitting was adjusted during the random selection of 20–20% of the total data. The training on four-fold data and the testing on the one-fold data are shown in Table 4.

In Table 4, there are 19 types of characters that have their fold-by-fold varying accuracy. The mean of all folds is given. The overall or mean of each fold and the mean of all folds are given in the last row. We can see that the Y character has a significant or the highest accuracy rate (95.40%) of validation compared to other characters. This may be due to its almost entirely different structure from other characters. The other highest accuracy is of the G character with 95.06%, which is almost equal to the highest with a slight difference. However, these two characters have a more than 95% recognition accuracy, and no other character is nearer to 95. The other characters have a range of accuracies from 81 to 90%. The least accurate M character is 62.08, and it varies in five folds from 53 to 74%. Therefore, we can say that M matches with other characters, and for this character recognition, we

**Table 4  Five-character dataset accuracy (%) and F1-score with five-fold text recognition based testing on the trained CNNs.**

| Character | Accuracy (%) | | | F1-measure (%) | | Accuracy (%) | F1-measure (%) |
|---|---|---|---|---|---|---|---|
| | Fold 1 | Fold 2 | Fold 3 | Fold 4 | Fold 5 | 5-Fold Mean | 5-Fold Mean |
| 2 | 87.23 | 83.33 | 89.63 | 84.21 | 83.14 | 84.48 | 86.772 |
| 3 | 87.76 | 75.51 | 87.75 | 90.323 | 89.32 | 86.12 | 86.0792 |
| 4 | 84.31 | 88.46 | 90.196 | 91.089 | 90.385 | 89.06 | 89.4066 |
| 5 | 84.31 | 80.39 | 90.00 | 90.566 | 84.84 | 86.56 | 85.2644 |
| 6 | 86.95 | 76.59 | 82.61 | 87.50 | 82.22 | 87.58 | 85.2164 |
| 7 | 89.36 | 87.23 | 86.95 | 86.957 | 88.636 | 86.68 | 87.3026 |
| 8 | 89.58 | 79.16 | 91.66 | 93.47 | 89.362 | 87.49 | 89.5418 |
| B | 81.81 | 73.33 | 97.72 | 86.04 | 90.90 | 85.03 | 87.7406 |
| C | 87.23 | 79.16 | 85.10 | 82.60 | 80.0 | 82.64 | 81.0632 |
| D | 91.30 | 78.26 | 91.30 | 87.91 | 88.66 | 88.67 | 86.7954 |
| E | 62.79 | 79.54 | 79.07 | 85.41 | 81.928 | 78.73 | 79.4416 |
| F | 92.00 | 84.00 | 93.87 | 93.069 | 82.47 | 89.1 | 87.5008 |
| G | 95.83 | 91.83 | 100 | 95.833 | 94.73 | 95.06 | 94.522 |
| M | 64.00 | 56.00 | 53.061 | 70.47 | 67.34 | 62.08 | 63.8372 |
| N | 81.40 | 79.07 | 87.59 | 79.04 | 78.65 | 81.43 | 77.8656 |
| P | 97.78 | 78.26 | 82.22 | 91.67 | 98.87 | 90.34 | 92.0304 |
| W | 95.24 | 83.72 | 90.47 | 96.66 | 87.50 | 90.55 | 91.3156 |
| X | 89.58 | 87.50 | 82.97 | 87.23 | 82.105 | 85.68 | 86.067 |
| Y | 93.02 | 95.45 | 97.67 | 95.43 | 95.349 | 95.40 | 95.8234 |
| **Overall** | **86.14** | **80.77** | **87.24** | **88.183** | **86.1265** | **85.52** | **85.9782** |

may need to concentrate on structural polishing for M input characters. To prevent CAPTCHA from breaking further complex designs among machines and making it easy for humans to do so, the other characters that achieve higher results need a high angle and structural change to not break with any machine learning model. This complex structure may be improved from other fine-tuning of a CNN, increasing or decreasing the skipping connection. The accuracy value can also improve. The other four-character dataset is essential because it has 32 characters and more images. This five-character dataset's lower accuracy may also be due to little data and less training. The other character recognition studies have higher accuracy rates on similar datasets, but they might be less confident than the proposed study due to an unbiased validation method. For further validation, precision and recall-based F1-Score for all five folds mean are shown in Table 4; the Y character again received the highest value of F1-measure with 95.82%. Using the proposed method again validates the 'Y' character as the most promisingly broken character. The second highest accuracy gaining character 'G' got the second-highest F1-score (94.522%) among all 19 characters. The overall mean F1-Score of all 5-folds is 85.97% that is more than overall accuracy. However, F1-Score is the harmonic mean of precision and recall, wherein this regard, it could be more suitable than the accuracy as it covers the class balancing issue between all categories. Therefore, in terms of F1-Score, the proposed study could

be considered a more robust approach. The four-character dataset recognition results are discussed in the next section.

### Four-character dataset ($d_2$)

The four-character dataset has a higher frequency of each character than the five-character dataset, and the number of characters is also higher. The same five-fold splits were performed on this dataset characters as well. After applying the five-folds, the number of characters in each fold was 7607, 7624, 7602, 7617, and 7595, respectively, and the remaining images from the 38,045 images of individual characters were adjusted into the training sets of each fold. The results of each character w.r.t each fold and the overall mean are given in Table 5.

From Table 5, it can be observed that almost every character was recognized with 99% accuracy. The highest accuracy of character D was 99.92 and remained 100% in the four-folds. Only one fold showed a 99.57% accuracy. From this point, we can state that the proposed study removed bias, if there was any, from the dataset by doing splits. Therefore, it is necessary to make folds in a deep learning network. Most studies use a 1-fold approach only. The 1-fold approach is at high risk. It is also essential that character M achieved the lowest accuracy in the case of the five-character CAPTCHA. In this four-character CAPTCHA, 98.58% was accurately recognized. Therefore, we can say that the structural morphology of M in the five-character CAPTCHA better avoids any CAPTCHA solver method. If we look at the F1-Score in Table 5, all character's recognition values range from 97 to 99%. However, the variation in all folds results remains almost the same as Folds accuracies. The mean F1-Scores against each character validate the confidence of the proposed method and the breaking of each type of character. The class balance issue in 32 types of classes is the big issue that could make less confident to the proposed method accuracy. However, the F1-Score is discussed and added in Table 5 that cross-validates the performance of the proposed study. The highest results show that this four-character CAPTCHA is at high risk, and line intersection, word joining, and correlation may break, preventing the CAPTCHA from breaking. Many approaches have been proposed to recognize the CAPTCHA, and most of them have used a conventional structure. The proposed study has used a more confident validation approach with multi-aspect feature extraction. Therefore, it can be used as a more promising approach to break CAPTCHA images and test the CAPTCHA design made by CAPTCHA designers. In this way, CAPTCHA designs can be protected against new approaches to deep learning. The graphical illustration of validation accuracy and the losses for both datasets on all folds is shown in Fig. 5.

The five- and four-character CAPTCHA fold validation losses and accuracies are shown. It can be observed that the all folds of the five-character CAPTCHA reached close to 90%, and only the 2nd fold value remained at 80.77%. It is also important to state that, in this fold, there were cases that may not be covered in other deep learning approaches, and their results remain at risk. Similarly, a four-character CAPTCHA with a greater number of samples and less complex characters should not be used, as it can break easily compared to the five-character CAPTCHA. CAPTCHA-recognition-based studies have

**Table 5  Four-character dataset accuracy (%) and F1-score with five-fold text recognition based testing on the trained CNNs.**

| Character | Accuracy (%) | | | | | Accuracy (%) | F1-measure (%) |
| | Fold 1 | Fold 2 | Fold 3 | Fold 4 | Fold 5 | 5-Fold Mean | 5-Fold Mean |
|---|---|---|---|---|---|---|---|
| 2 | 97.84 | 99.14 | 99.57 | 98.92 | 99.13 | 98.79 | 98.923 |
| 3 | 97.02 | 94.92 | 98.72 | 97.403 | 97.204 | 96.52 | 97.5056 |
| 4 | 97.87 | 97.46 | 99.15 | 98.934 | 98.526 | 98.55 | 98.4708 |
| 5 | 98.76 | 98.76 | 99.17 | 97.97 | 98.144 | 99.01 | 98.0812 |
| 6 | 100 | 95.65 | 99.56 | 99.346 | 99.127 | 98.69 | 98.947 |
| 7 | 98.80 | 99.60 | 99.19 | 99.203 | 98.603 | 99.36 | 98.9624 |
| 8 | 99.15 | 98.72 | 97.42 | 98.283 | 98.073 | 98.29 | 98.1656 |
| 9 | 98.85 | 96.55 | 98.08 | 98.092 | 99.617 | 98.39 | 98.4258 |
| A | 97.85 | 98.71 | 99.13 | 98.712 | 97.645 | 98.54 | 98.2034 |
| B | 99.57 | 96.59 | 98.72 | 97.89 | 96.567 | 97.95 | 97.912 |
| C | 99.58 | 98.75 | 99.16 | 99.379 | 99.374 | 99.25 | 99.334 |
| D | 100 | 100 | 100 | 99.787 | 99.153 | 99.92 | 99.6612 |
| E | 99.18 | 97.57 | 100 | 98.994 | 98.374 | 98.94 | 98.6188 |
| F | 98.69 | 98.26 | 100 | 98.253 | 98.253 | 98.52 | 98.3076 |
| G | 98.76 | 97.93 | 100 | 98.319 | 98.551 | 98.43 | 98.7944 |
| H | 99.58 | 97.90 | 100 | 98.347 | 99.371 | 99.33 | 99.1232 |
| J | 100 | 98.72 | 99.57 | 99.788 | 99.574 | 99.66 | 99.4458 |
| K | 99.15 | 99.58 | 100 | 99.156 | 99.371 | 99.58 | 99.1606 |
| L | 97.41 | 98.28 | 100 | 99.355 | 99.352 | 98.79 | 99.1344 |
| M | 99.16 | 96.23 | 99.16 | 99.17 | 99.532 | 98.58 | 98.9816 |
| N | 99.58 | 97.10 | 99.17 | 99.793 | 98.755 | 98.83 | 98.652 |
| P | 98.35 | 97.94 | 98.77 | 98.347 | 96.881 | 97.86 | 97.8568 |
| Q | 100 | 100 | 99.58 | 99.576 | 99.787 | 99.75 | 99.7456 |
| R | 99.58 | 99.17 | 99.17 | 99.174 | 98.319 | 99.00 | 99.0834 |
| S | 98.75 | 99.58 | 100 | 99.583 | 99.156 | 99.42 | 99.4118 |
| T | 97.47 | 97.90 | 98.73 | 98.305 | 98.312 | 97.98 | 99.558 |
| U | 100 | 97.43 | 99.57 | 99.134 | 98.925 | 98.80 | 99.1794 |
| V | 100 | 98.67 | 98.67 | 99.332 | 98.441 | 98.47 | 98.8488 |
| W | 100 | 100 | 100 | 99.376 | 99.167 | 99.67 | 99.418 |
| X | 99.15 | 97.46 | 100 | 99.573 | 99.788 | 99.15 | 99.3174 |
| Y | 97.90 | 98.33 | 98.74 | 98.156 | 99.371 | 98.66 | 98.7866 |
| Z | 99.17 | 98.75 | 99.16 | 98.965 | 99.163 | 99.16 | 99.0832 |
| **Overall** | **98.97** | **98.18** | **99.32** | **98.894** | **98.737** | **98.82** | **98.846** |

used self-generated or augmented datasets to propose CAPTCHA solvers. Therefore, the number of images, their spatial resolution sizes and styles, and other results have become incomparable. The proposed study mainly focuses on a better validation technique using deep learning with multi-aspect feature *via* skipping connections in a CNN. With some character-matching studies, we performed a comparison to make the proposed study more reliable.

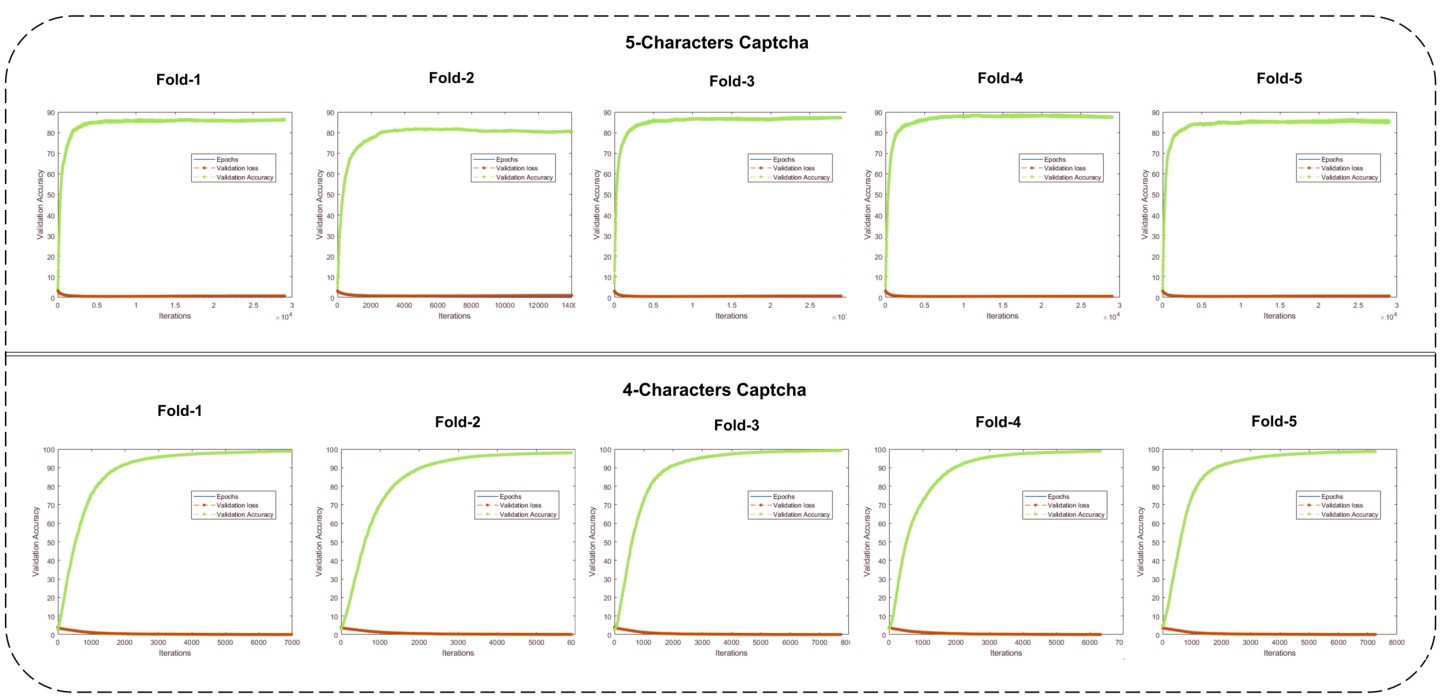

**Figure 5** The validation loss and validation accuracy graphs are shown for each fold of the CNN (row 1: five-character CAPTCHA; row 2: four-character CAPTCHA).

**Table 6** Comparison of proposed study based five and four-character datasets' with state-of-the-art methods.

| References | No. of characters | Method | Results |
|---|---|---|---|
| *Du et al. (2017)* | 6 | Faster R-CNN | Accuracy = 98.5% |
| | 4 | | Accuracy = 97.8% |
| | 5 | | Accuracy = 97.5% |
| *Chen et al. (2019)* | 4 | Selective D-CNN | Success rate = 95.4% |
| *Bostik et al. (2021)* | Different | CNN | Accuracy = 80% |
| *Bostik & Klecka (2018)* | Different | KNN | Precision = 98.99% |
| | | SVN | 99.80% |
| | | Feed forward-Net | 98.79% |
| Proposed Study | 4 | Skip-CNN with 5-Fold Validation | Accuracy = 98.82% |
| | 5 | – | Accuracy = 85.52% |

In Table 6, we can see that various studies have used different numbers of characters with self-collected and generated datasets, and comparisons have been made. Some studies have considered the number of dataset characters. Accuracy is not comparable, as it uses the five-fold validation method, and the others only used 1-fold. Therefore, the proposed study outperforms in each aspect, in terms of the proposed CNN framework and its validation scheme.

## CONCLUSION

The proposed study uses a different approach to deep learning to solve CAPTCHA problems. It proposed a skip-CNN connection network to break text-based CAPTCHAs. Two CAPTCHA datasets are discussed and evaluated character by character. The proposed study is confident to report results, as it removed biases (if any) in datasets using a five-fold validation method. The results are also improved as compared to previous studies. The reported higher results claim that these CAPTCHA designs are at high risk, as any malicious attack can break them on the web. Therefore, the proposed CNN could test CAPTCHA designs to solve them more confidently in real-time. Furthermore, the proposed study has used the publicly available datasets to perform training and testing on them, making it a more robust approach to solve text-based CAPTCHA's.

Many studies have used deep learning to break CAPTCHAs, as they have focused on the need to design CAPTCHAs that do not consume user time and resist CAPTCHA solvers. It would make our web systems more secure against malicious attacks. However, in the future, the data augmentation methods and more robust data creation methods can be applied on CAPTCHA datasets where intersecting line-based CAPTCHA's are more challenging to break that can be used. Similarly, the other local languages based CAPTCHAs also can be solved using similar DL models.

### Funding
The authors received no funding for this work.

### Competing Interests
Shida Lu is employed by State Grid Information & Communication Company, SMEPC, China.

Kai Huang is employed by Shanghai Shineenergy Information Technology Development Co., Ltd., China

### Author Contributions
- Shida Lu and Kai Huang conceived and designed the experiments, performed the experiments, analyzed the data, performed the computation work, prepared figures and/or tables, authored or reviewed drafts of the paper, and approved the final draft.
- Talha Meraj conceived and designed the experiments, performed the experiments, performed the computation work, prepared figures and/or tables, authored or reviewed drafts of the paper, and approved the final draft.
- Hafiz Tayyab Rauf conceived and designed the experiments, analyzed the data, performed the computation work, prepared figures and/or tables, authored or reviewed drafts of the paper, and approved the final draft.

### Data Availability
The MATLAB code is available in the Supplemental File. The data is available at Kaggle: https://www.kaggle.com/genesis16/captcha-4-letter and https://www.kaggle.com/fournierp/captcha-version-2-images.

## Supplemental Information

Supplemental information for this article can be found online at http://dx.doi.org/10.7717/peerj-cs.879#supplemental-information.

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
