# Peer review of "A novel CAPTCHA solver framework using deep skipping Convolutional Neural Networks"

_PeerJ Computer Science, doi:10.7717/peerj-cs.879_

## Round 0.1 · original submission · Major Revisions

Please revise your manuscript in accordance to the reviewers' comments. Although the reviewers pointed out minor issues with the manuscript, these issues combined warrant a Major Revision of your work.

Reviewer 1 ·

Basic reporting

This research primarily used deep learning methods to break state-of-the-art CAPTCHA codes. Overall the idea is good and the study is organized, however, there are still some improvements that need to be done. Please use the following comments and revise the manuscript accordingly.

Experimental design

1. The proposed approach used two different types of datasets on the same model. How can the same prediction model predict the different types of captchas? Please elaborate.
2. The previous studies have used dataset generation, where the proposed study used only a public study dataset. Why?

Validity of the findings

1. It is said that 5-folds have been used, where no detail of each fold is given that how it is calculated. Please evaluate it in terms of table or diagram etc.
2. The 'conclusions' are a crucial component of the paper. It should complement the 'abstract' and is normally used by experts to value the paper's engineering content. In general, it should sum up the most important outcomes of the paper. It should simply provide critical facts and figures achieved in this paper for supporting the claims.

Additional comments

1. The proposed study has used a proposed CNN, where in previous studies, most of the researchers have used similar CNNs. How the proposed CNN is different and novel as compared to previous studies? Clearly mention and highlight.
2. The abstract is too general and not prepared objectively. It should briefly highlight the paper's novelty as the main problem, how it has been resolved, and where the novelty lies?
3. The existing literature should be classified and systematically reviewed instead of being independently introduced one by one.
4. It seems that Skip connection is used in the proposed CNN; how does it work differently? No detail of this part is given. Please evaluate it specifically in some separate sections to get into depth.

Reviewer 2 ·

Basic reporting

The author proposed deep learnig method for captcha breaking security. The paper technically sounds, however, there are still several issues needs to be fixed.

In the Literature review, the table shows only CNN-based approaches in 2021, mainly where if the table is created, it must highlight different aspects of studies from different years.

The proposed study has used text-based CAPTCHA images that contain multiple characters where single character recognition results and predictions are shown? How is the segmentation processed? Are previous studies similarly doing segmentation?

What is the reason for choosing these specific datasets? When researchers using their own generated dataset, that is more in several images as well.

What does mean by 5-fold validation in CNN? Please clearly make a separate section, also mention what its significance?

In the comparison section, in the table, it is said “different” type of characters! What does it mean by that? Please elaborate on the results and comparison section.

There are some typos and grammatical errors in the manuscript. It is strongly suggested that the whole work to be carefully checked by someone has expertise in technical English writing.

Key contribution and novelty has not been detailed in manuscript. Please include it in the introduction section

What are the limitations of the related works

Are there any limitations of this carried out study?

How to select and optimize the user-defined parameters in the proposed model?

There are quite a few abbreviations are used in the manuscript. It is suggested to use a table to host all the frequently used abbreviations with their descriptions to improve the readability

Explain the evaluation metrics and justify why those evaluation metrics are used?

It seems that the authors used images of equations, please use editable equation format.

The Related Works section is also fair, yet the criteria behind the selection of the works described should be explained.

Experimental design

please see above

Validity of the findings

please see above

Additional comments

please see above

Reviewer 3 ·

Basic reporting

What are the research gaps in previous studies, and what are your contributions? Have not to mention clearly.
Does the proposed CNN have a skip connection? What is that? How is it working? Why do you use it?
How is the segmentation step processed? It is missing in the manuscript. It should be separately discussed.

Experimental design

The substantial contributions should be highlighted and discussed. The results and comparative analysis should be discussed in detail.
Every time a method/formula is used for something, it needs to be justified by either (a) prior work showing the superiority of this method, or (b) by your experiments showing its advantage over prior work methods - comparison is needed, or (c) formal proof of optimality. Please consider more prior works.

Validity of the findings

The data is not described. Proper data description should contain the number of data items, number of parameters, distribution analysis of parameters, and target parameter for classification.
Figure resolutions are bad. The text in figures became unreadible due to digitizing. Some figures have black background. Why the background is black for them?
8. The tables are too long. What te authors are trying to show in the tables are not clear?
9. Introduction contains too many citations. Please keep them at literature review. Authors are not supposed to explain the results in the introduction part.

---

## Round 0.2 · Minor Revisions

Please revise the article accordingly to reviewers' comments.

Reviewer 1 ·

Basic reporting

The entire paper has been written perfectly.
Literature is sufficient.
The organization of the paper is good.

Experimental design

Experiments are valid.
Results are presented systematically.
Moreover, the quality of the paper has been improved.

Validity of the findings

All findings are well written and presented.

Additional comments

The paper can be accepted.

Reviewer 2 ·

Basic reporting

The manuscript is relatively improved. However, I still think the manuscript needs reorganization, and contribution needs to be more precise. Please see the following comments, and hence, I would suggest one more round of revision.

Experimental design

*Please fix Table 1 position; It should be in line with the LR section.

*Similar Issue with Table 3, also please rewrite Table 3 caption.

Validity of the findings

*Table 5 borders are out of page width; please fix this as well.

*Would you please add two-three more points on the contribution of the mathdology?

Additional comments

See above

Reviewer 3 ·

Basic reporting

I recommend to accept the current version

Experimental design

I recommend to accept the current version

Validity of the findings

I recommend to accept the current version

Additional comments

I recommend to accept the current version

Reviewer 4 ·

Basic reporting

The manuscript is written in poor English that not suitable for publish in this journal until it be improved to ensure that audiences can clearly understand the text. There are a lot of short sentences that lack context.

Experimental design

Not bad.

Validity of the findings

Not bad.

Additional comments

Not bad.

Reviewer 5 ·

Basic reporting

Initially, I would like to appreciate the authors efforts for their interesting work. The work is a timely one since the deep learning based CAPTCHA breaking has been gained momentum within the recent years. Generally, the work can be evaluated as good and worth reading one, however there are several issues which should be taken into account before it could be accepted for publication.
1. The text, throughout the manuscript and more specifically in the abstract, introduction and literature review is not integrated, well-written. It mostly seems several short, and in some cases unrelated sentences without any coherence. So, the text does not seem natural and should be revised, maybe by a native person.
2. Some of the references in Introduction section are not necessary, since those are related to obvious and common information not something specific and important. For example, in line 49, "Azad and Jain (2013). CAPTCHA can be used for authentication in login forms with various web credentials". Instead, the authors can refer to some of the important survey papers (such as [1,2]) in the this field for all of such information.

[1]Roshanbin, N., & Miller, J. (2013). A survey and analysis of current captcha approaches. Journal of Web Engineering, 001-040.
[2]Xu, X., Liu, L., & Li, B. (2020). A survey of CAPTCHA technologies to distinguish between human and computer. Neurocomputing, 408, 292-307.

3. In the Literature Review section, specifically lines 101 to 109, the organization of the information is poor and need to be revised to be more clear and readable.
4. Generally, the Literature Review section is not well-organized and both its text and structure should be improved. For example, the authors may introduce some of classical CPATCHA breaking methods and elaborate on their deficiency. Then, they can speak about benefits of deep learning based methods and mention the notable works in this domain, either chronologically or based on underlying approach.
5. Some of the tables are not appropriately located in the manuscript. It may be due to inconsistency in Latex but anyway the issue should be managed.
6. Figure 1 should be revised. There are some problems with the figure. For example, the connection between different sections is not illustrated appropriately. This is not clear, for example, how input image will be passed to the next step. Also, the circle in the upper section of the figure is not large enough to fit the "preprocessing" term.
7. The section which is entitled "Comparison" seems to be unnecessary since its information could be presented in the discussion section.
8. The authors are encouraged to tell about the possible future works in the field.
9. The major motivation of the study should be clearly stated, may be in a separate section. What is the advantage of this study and what it brings to the community? What problem(s) this work is intended to solve and what is new in this research?
10. Since the most of today's CAPTCHAs are image-based, the authors should clarify why such works are still useful and worth considering. Can such research come in handy for other applications?

Experimental design

The experiment is well-designed and conducted, so it seems appropriate. It is suggested that the authors speak about the efficiency of the proposed method for text-based CAPTCHAs in other languages, either experimentally or theoretically.

Validity of the findings

The findings are acceptable and worth considering.

Additional comments

If the authors could revise their manuscript appropriately and according to the comments, it could be accepted for publication.

---

## Round 0.3 · Minor Revisions

Please revise (or rebut) the article according to the reviewer's 4 comments especially on the concerns on the benchmark models. It was commented that the references of the investigated models as benchmarks are not convincing.

Reviewer 2 ·

Basic reporting

Accept

Experimental design

Accept

Validity of the findings

Accept

Additional comments

Accept

Reviewer 4 ·

Basic reporting

Although the author has improved the manuscript, I still have some concerns on the benchmark models. I check the references of the investigated models, and find the benchmark models are not convincing. Here, I suggest authors to compare the proposed algorithm with some recent state-of-art algorithms from some top tier journals (IEEE Transactions on Information Forensics and Security, Computers & Security) in computer security.

Experimental design

Not bad.

Validity of the findings

Authors have to compare the proposed algorithms with some state-of-art methods in some top journals.
I find many work on the CAPTCHA solver framework from the google scholar as below.
1. Alqahtani, Fatmah H., and Fawaz A. Alsulaiman. "Is image-based CAPTCHA secure against attacks based on machine learning? An experimental study." Computers & Security 88 (2020): 101635.
2. Ouyang, Zhiyou, et al. "A cloud endpoint coordinating CAPTCHA based on multi-view stacking ensemble." Computers & Security 103 (2021): 102178.
3. Osadchy, Margarita, et al. "No bot expects the DeepCAPTCHA! Introducing immutable adversarial examples, with applications to CAPTCHA generation." IEEE Transactions on Information Forensics and Security 12.11 (2017): 2640-2653.
4. Shi, Chenghui, et al. "Adversarial captchas." IEEE Transactions on Cybernetics (2021).

Additional comments

The current experiment is not convincing.

Reviewer 5 ·

Basic reporting

The reviewer would like to appreciate the authors' work and effort in revising the manuscript. So, the manuscript has been improved and can be published in its current form.

Experimental design

The reviewer would like to appreciate the authors' work and effort in revising the manuscript. So, the manuscript has been improved and can be published in its current form.

Validity of the findings

The reviewer would like to appreciate the authors' work and effort in revising the manuscript. So, the manuscript has been improved and can be published in its current form.

---

## Round 0.4 · Minor Revisions

1) Please show examples of color images. In your current article, only grayscale images are demonstrated.

2) Please explain in your captions of figure and title of table, why are these tables or figures necessary in your paper? What are the purposes and what are the message you want to deliver via these figures and tables?

3) The metric "Accuracy" might not be sufficient to judge the performance of the model holistically. Please enhance the result analysis part of your paper.

Reviewer 4 ·

Basic reporting

The manuscript can be accepted.

Experimental design

I have no more comment.

Validity of the findings

I have no more comment.

Additional comments

I have no more comment.

---

## Round 0.5 · accepted · Accept

All concerns have been addressed. A decision of acceptance has been made. Congratulations!